# OpenReview forum: "Unified Safe In-context Image Generation in Multimodal Diffusion Transformers via Restricting Unsafe Information Flows"
_ICML.cc/2026/Conference — ICML 2026 regular_

### Official Review · Reviewer_UCbu · 2026-03-09

**Soundness:** 2
**Presentation:** 3
**Significance:** 2
**Originality:** 2
**Overall Recommendation:** 3
**Confidence:** 4

**Summary:**

This paper proposes the Unified Visual Safety Regulator (UVR), a training-free framework designed to mitigate the generation of unsafe content. The core of the approach lies in an in-depth analysis of attention dynamics within the MM-Attn mechanism, identifying a task-independent "start-up stage" where harmful semantics emerge and can be localized. By pre-collecting "unsafe anchors" offline, UVR performs patch-level localization and subsequently restricts harmful information flow through targeted attention modulation and noise injection during inference.

**Compliance With Llm Reviewing Policy:**

Affirmed.

**Ethical Review Concerns:**

This article contains some uncensored pornographic images.

**Ethical Review Flag:**

Flag this paper for an ethics review.

**Ethics Expertise Needed:**

["Other Expertise"]

**Final Justification:**

I thank the authors for their detailed response. The clarifications have resolved my concerns regarding Figure 3, efficiency, and image artifacts. However, I remain unconvinced regarding the framework's practical generalization and its dependency on external models. Specifically, the sensitive hyperparameter still requires manual, concept-specific selection depending on whether a concept is "general" or "concrete", which contradicts the "unified" claim and hinders its utility in real-world scenarios where the threat type is often unknown and various. Furthermore, the evaluation scope remains narrow for a framework claiming broad applicability across diverse safety risks. In summary, I appreciate the technical improvements and will change the score to 3, but I maintain a rejection inclination as these core limitations regarding practical deployment and open-set robustness persist.

**Key Questions For Authors:**

- What is the specific base model used for the analysis in Figure 3? Is this pattern consistent across other MM-DiT variants or is it model-specific? Also, please clarify the meaning of the translucent lines (e.g., the orange lines) in Figure 3.
- In a real-world black-box generation scenario where the system does not know which unsafe concept the user is targeting, how does UVR adaptively and dynamically determine the sensitive hyperparameters (e.g., $\tau%)?
- Does UVR possess "zero-shot" defense capabilities to intercept novel or obscured harmful content that was not represented in the offline anchor collection phase?
- What is the overall computational efficiency of the framework? Specifically, how much inference latency and VRAM overhead does the patch-level similarity matching and connected component analysis introduce compared to standard baselines?

**Limitations:**

No, from my perspective. While the authors demonstrate the effectiveness of their method, they should more explicitly discuss the limitations regarding hyperparameter sensitivity, the potential for visual artifacts in suppressed regions, and the reliance on pre-defined anchor sets for novel safety threats.

**Strengths And Weaknesses:**

# Strength
- This paper considers a notable topic that is highly relevant to the responsible deployment of large-scale generative models, especially given the rising popularity of DiT-based architectures.
- The analysis of attention dynamics from the perspective of information flow provides a fresh and insightful viewpoint for understanding how semantics are incorporated into output tokens.
- The paper is generally well-written with a smooth narrative and clear logical flow.
# Weakness
- The core localization mechanism relies on external models (e.g., Grounded-SAM) for visual grounding and anchor collection. This creates a dependency where the safety ceiling is limited by these external models. Furthermore, if such powerful external detectors are available, one could arguably implement a simpler input/output filtering pipeline, which significantly diminishes the fundamental necessity of the proposed complex internal regulator.
- While the paper claims to be a "task-agnostic" and "unified" framework, the experimental details reveal that different tasks and concepts require distinct, manually selected hyperparameters (e.g., $\tau$). This reliance on heuristic per-concept tuning raises serious concerns about its practical utility in real-world, open-set scenarios.
- The attention manipulation and noise injection appear somewhat destructive. Qualitative results suggest that the regions where unsafe concepts are suppressed often exhibit noticeable artifacts, blurring, or unrealistic blending, which affects overall image fidelity.
- The experiments only cover four specific types of unsafe concepts (Nudity, Pikachu, Blood, Weapon). For a method claiming to be "unified," this scope is insufficient to demonstrate broad generalizability across the diverse landscape of harmful content.
- Several figures suffer from crowded layouts and a lack of sufficient legends/labels, hindering readability. Additionally, there is a citation error in Section 3.1, where Figure 3 is incorrectly referred to as Figure 2.

---

> ### Author Rebuttal · Authors · 2026-03-31
>
> We sincerely thank the reviewer for the careful evaluation and constructive feedback. We are glad that the motivation and presentation of our work are considered clear and insightful. Below, we provide detailed clarifications addressing each concern.
>
> # R1: Role of Grounded-SAM and Necessity of Internal Regulation (W1)
> Grounded-SAM is applicable only to fully generated images for anchor collection, whereas UVR performs **step-wise localization via MMDiT’s internal attention**, enabling **stronger generalization** that external detectors cannot provide.
> Besides, input/output filtering (LLaMAGuard;LLaVAGuard[1]) is **orthogonal** to internal regulation (ESD;EA;ours), which improves safety while preserving user experience through high-quality outputs.
>
> [1]LlavaGuard: An Open VLM-based Framework for Safeguarding Vision Datasets and Models, ICML25
>
> # R2: Clarifying *Task-agnostic/Unified* and Practical Utility (W2)
> We use **unified** to denote that UVR provides a *single framework applicable to both T2I and I2I tasks* under a consistent information-flow perspective; **task-agnostic** refers to this shared formulation. Regarding **practical utility**, we further conduct a parameter study of $\tau$ (R3) and demonstrate scalability to a broader scope (bias suppression in R5).
>
> # R3: Parameter Study of $\tau$ (W2/Q2)
> As shown in https://ibb.co/N6LccrRR, UVR preserves image quality over a broad range (0.3–0.65), while safety effectiveness decreases monotonically with $\tau$ (↓).
> The selection of $\tau$ is mainly related to differences in the model’s internal representations of concepts; for instance, **general concepts (e.g., nudity, blood)** associating with multiple attributes tend to yield higher attention scores and favor larger $\tau$ (0.5–0.6), whereas **concrete concepts (Pikachu, weapon)** work well with lower $\tau$ (0.3–0.4).
>
> # R4: Clarification about Artifacts (W3)
> Qualitative results (Main.Fig.1, Fig.6) show that UVR does not introduce noticeable artifacts when suppressing unsafe concepts. Ablation results (Fig.15-Col3) show that *checkerboard artifacts* only arise *when suppression is applied at all denoising steps*, whereas UVR adopts adaptive step selection, which avoids this issue (Col5). The *camouflage-like patterns* observed for 'weapon' reflect model's **default generative behaviors for such scenes** rather than artifacts.
>
> We will clarify this in Sec.6 of the manuscript.
>
> # R5: Broad Generalizability beyond Four Concepts (W4)
> To validate scalability, we extend UVR to mitigate gender bias in T2I generation. Specifically, we perform gender-specific anchor collection and modulate information flow to balance underrepresented groups.
> We observe that **attention score residuals** between groups reveal bias—e.g., 0.119 for Nurse (female-biased), –0.107 for Software Engineer (male-biased), and 0.014 for Doctor (unbiased)—enabling targeted regulation via information flow modulation with auxiliary text vectors.
> As shown in https://ibb.co/VYhyRjj9, on MMDT benchmark[1] (62/13 concepts for Occupation/Education), UVR **reduces Group Unfairness from –0.429/–0.556 to –0.048/–0.071 (closer to 0 is better)**, demonstrating strong scalability and generalization beyond the limited concept set.
>
> [1]MMDT: Decoding the Trustworthiness and Safety of Multimodal Foundation Models, ICLR25
>
> # R6: Details of Fig.3 & Generalizability of Attention Dynamics (Q1)
> Fig.3 is based on FLUX.1-dev (T2I) and FLUX.1-kontext (I2I). The **translucent** lines denote non-dominant information flows, which carry relatively limited information across layers and denoising steps, while non-translucent lines denote dominant flows. Regarding **generalizability**, we observe similar attention dynamics in another MMDiT variant, FLUX.1-schnell. Anchors collected on -dev also effectively regulate the “nude” concept on -schnell. As shown in https://ibb.co/bg98K7Vr, the harm ratio is reduced from 42.89% to 14.69%, while CLIP scores remain stable (31.50 → 31.51).
>
> We will *improve the readability of Fig.3, refine the layout of Fig.4, and correct the citation error in Sec.3.1.*
>
> # R7: Zero-Shot Defense against Novel or Obscured Harmful Content (Q3)
> UVR can generalize to intercept *semantically related but unseen* unsafe concepts.
> For example, anchors derived from “female breasts” effectively suppress other nude concepts (“genitalia/buttocks”, Fig.11), as well as stylistic variations (realistic and anime styles, Fig.12-Row1,3), which is not included during anchor collection.
>
> # R8: Computational Efficiency (Q4)
> We compare the latency and VRAM overhead of UVR with baselines in https://ibb.co/zhknXSPG. All metrics are evaluated on the same platform. UVR incurs slightly higher memory overhead than Flux, and achieves lower latency than SLD as it intervenes only in early denoising steps (vs. SLD’s double forward passes per step).
>
> *Finally*, we appreciate the reviewer’s suggestions and will revise the presentation of unsafe images with appropriate masking.

---

> > ### Author Rebuttal · Reviewer_UCbu · 2026-04-03
> >
> > I thank the authors for their detailed response. The clarifications have resolved my concerns regarding Figure 3, efficiency, and image artifacts. However, I remain unconvinced regarding the framework's practical generalization and its dependency on external models. Specifically, the sensitive hyperparameter still requires manual, concept-specific selection depending on whether a concept is "general" or "concrete", which contradicts the "unified" claim and hinders its utility in real-world scenarios where the threat type is often unknown and various. Furthermore, the evaluation scope remains narrow for a framework claiming broad applicability across diverse safety risks. In summary, I appreciate the technical improvements and will change the score to 3, but I maintain a rejection inclination as these core limitations regarding practical deployment and open-set robustness persist.

---

> > > ### Author Response · Authors · 2026-04-03
> > >
> > > We thank the reviewer for acknowledging the clarifications provided in our rebuttal.
> > > We provide further clarifications on the remaining concerns; we believe these additional explanations help better characterize the intended scope and practical applicability of our framework.
> > >
> > > **C1: Practical generalization concerns under unknown threats.**
> > > - We clarify that, in real-world deployment, defending against unknown threats remains an inherently open problem and *there is currently no complete solution that can proactively cover all emerging risks*. In practice, many safety issues emerge as **zero-day threats**, where **novel misuse patterns appear in the wild only after deployment**. A recent example is the misuse of OpenAI's GPT-Image for creating content in the style of Studio Ghibli, in both T2I and I2I settings.
> > > - In this context, UVR operates as an inference-time mechanism that **enables rapid mitigation in both T2I and I2I** once a threat is identified, without requiring retraining.
> > >
> > > **C2: Generalization concerns regarding hyperparameter selection.**
> > > - We argue that concept-specific $\tau$ for attention modulation is both **reasonable and necessary**, as MMDiTs **exhibit distinct attention patterns across different concept types**. This does not weaken UVR’s generalizability; rather, it shows that UVR can adapt to diverse risk concepts under a unified suppression mechanism, without changing its core design.
> > > - As shown in R3, $\tau$ can be selected automatically via a simple probing strategy (we also provide the figures from R3 here: https://ibb.co/N6LccrRR), which can be summarized as follows:
> > >     - We scan $\tau \in [0.1, 0.9]$ in descending order and jointly evaluate (1) image quality via $\delta \text{CLIP}(\tau)$ relative to the base model, (2) harmfulness reduction via $\delta \text{HR}=\text{HR}(\tau+0.05)-\text{HR}(\tau)$.
> > >     - In practice, a stable region ($[0.3,0.65]$) already ensures negligible quality degradation ($\delta \text{CLIP}<0.4$), and an appropriate $\tau$ can be selected if the harmfulness reduction is significant ($\delta \text{HR}>6$).
> > > - Besides, we clarify that the distinction between general and concrete concepts in R3 is introduced to explain the underlying attention behaviors in MMDiTs, rather than to suggest heuristic manual tuning.
> > >
> > >
> > > **C3: Practical generalization concerns arising from dependency on external models**
> > > - We use Grounded-SAM as an external model to **automate anchor collection**, which is **effective within our current scope**. We appreciate the reviewer’s concern that its capability may limit UVR's performance on abstract or open-vocabulary concepts, since *such models mainly rely on region-level visual-text alignment and lack compositional reasoning over higher-level semantics*. Addressing this limitation remains an open challenge.
> > > - **Importantly, however, Grounded-SAM's limitation has negligible practical impact on generalization of UVR.** As shown in (R1 to Reviewer mau3), a small number of anchors (e.g., 29 anchors from 2 images) is already sufficient to achieve a strong safety-efficiency trade-off.
> > > Because the anchor collection cost is very low, even for novel or complex concepts it can be completed efficiently with limited human effort (e.g., within 2-5 minutes), **without affecting UVR’s main contributions in attention dynamics analysis and information flow modulation.**
> > > (Please refer to R1 to Reviewer mau3 for detailed results, as we are unable to include the tables here due to space limitations.)
> > >
> > > **C4: Practical generalization concerns arising from evaluation scope across diverse safety risks & UVR as a “unified” framework.**
> > > - Firstly, we clarify that UVR is termed a **“unified”** framework due to its applicability to both T2I and I2I settings under a shared mechanism of attention dynamics analysis and information flow modulation.
> > > - To further address the concern on **evaluation scope**, we further demonstrate UVR’s generalizability across concepts **including artistic style (Van Gogh) and celebrity (Taylor Swift) in both T2I and I2I settings**. Qualitative and quantitative results in https://ibb.co/5gfpL2gw demonstrate that UVR generalizes across safety scenarios. Specifically:
> > >     - Concept-specific anchors can accurately identify regions where the target concept emerges. For example, the attention score reaches 0.5825 in Taylor Swift-related generations, while only 0.1942 in safe images.
> > >     - We apply modulation on unsafe information flows using automatically determined $\tau$ (0.35 for Van Gogh and 0.5 for Taylor Swift). As illustrated in Tab.2, **UVR reduces Harm rate from 68.49/79.59 to 25.61/27.08**, while maintaining comparable CLIP scores (31.41/31.43 vs. 31.49 for FLUX.1-dev).
> > >
> > > ---
> > >
> > > Finally, we sincerely appreciate the reviewer’s time and effort in evaluating our work and for the constructive feedback. We would be happy to provide further clarification or discussion on any remaining concerns.
> > >
> > > Best regards,
> > >
> > > Authors

---

### Official Review · Reviewer_xnpP · 2026-03-12

**Soundness:** 3
**Presentation:** 4
**Significance:** 3
**Originality:** 4
**Overall Recommendation:** 4
**Confidence:** 4

**Summary:**

This paper proposes a security vulnerability solution for state-of-the-art multimodal diffusion, exemplified by FLUX.1. The authors point out that most existing security mechanisms are text-centric and designed for text-to-image tasks, leaving significant vulnerabilities in instruction-driven image-to-image editing. Through rigorous analysis of cross-modal attention mechanisms, the authors identify the "semantic initiation phase" in T2I and the "persistent interference phase" in I2I, discovering that harmful information manifests in early output image patches. Based on this, the authors propose a Unified Visual Security Modulator (UVR), a training-free, inference-time intervention method. UVR utilizes pre-collected anchor embeddings to locate insecure visual patches and suppresses insecure semantics by reducing specific cross-modal attention weights and injecting directional noise. Extensive experiments demonstrate that UVR effectively reduces security risks in both T2I and I2I tasks while maintaining image generation quality and editing fidelity.

**Compliance With Llm Reviewing Policy:**

Affirmed.

**Ethical Review Concerns:**

The authors have placed uncensored nude images in some of the images in this paper (e.g., Figure 6, 10, 11, 13), which may offend readers and cause mental harm to them. Although there is a warning at the beginning, I strongly recommend that the authors blur certain images.

**Ethical Review Flag:**

Flag this paper for an ethics review.

**Ethics Expertise Needed:**

["Inappropriate Potential Applications & Impact (e.g., human rights concerns)", "Responsible Research Practice (e.g., IRB, documentation, research ethics)", "Legal Compliance (e.g., EU AI Act, GDPR, copyright, terms of use)"]

**Final Justification:**

Thanks for the rebuttal. The author's explanation of Practical Scalability clarified my concerns, and the relevant experiments were conducted in FLUX.1-schnell. Thus I maintained my positive score.

**Key Questions For Authors:**

- How does the UVR framework scale to a massive number of diverse or highly abstract unsafe concepts? Could the anchor construction process be automated dynamically (leveraging Vision-Language Models)?
- The method relies on hard-coded thresholds for different concepts. How sensitive is the overall performance to these specific values, and is there a principled way to determine them without exhaustive empirical search?

**Limitations:**

Yes.

**Strengths And Weaknesses:**

## Strengths

- The author focuses the defense on the output, overcoming the limitations of prompt filtering and ddresses the overlooked I2I safety blind spot in MM-DiTs.
- The empirical analysis of multimodal attention dynamics is very solid, providing strong theoretical support for UVR design. The dual intervention mechanism of attention modulation and noise injection is logically rigorous.
- The method presented in the paper is training-free; it only needs to adjust the attention during the inference phase, making it simple and efficient.
- The experiments covered a variety of concepts, including nudity, copyrighted IP, and inappropriate objects. UVR achieved erasure rates of 91% and 77% in T2I and I2I tasks, respectively, while effectively preserving the original visual quality and editing fidelity. Facing modifier-based jailbreak attacks, UVR demonstrated good defensive robustness, significantly reducing the proportion of insecure images generated.

## Weaknesses

- The paper's ability to generalize in real-world scenarios is questionable. The paper's method heavily relies on a predefined library of "unsafe anchors" and manually set thresholds (0.6 for nudity and 0.35 for Pikachu), which raises concerns about the model's scalability.
- The noise injection mechanism heavily relies on the "early semantic initiation phase" assumption in the standard multi-step diffusion process. How will UVR perform on short-step or distilled DiT models with highly compressed temporal dynamics (such as FLUX-schnell)?
- The authors have placed uncensored nude images in some of the images in this paper (e.g., Figure 6, 10, 11, 13), which may offend readers and cause mental harm to them. This raises ethical concerns. Although there is a warning at the beginning, I strongly recommend that the authors blur certain images.

---

> ### Author Rebuttal · Authors · 2026-03-31
>
> We thank the reviewer for the time and effort in evaluating our work. We are encouraged by the positive feedback on our contributions and the soundness of the proposed method. Below, we provide clarifications and additional evidence addressing the raised concerns.
>
> # R1: Robustness of threshold $\tau$ and Practical Scalability (W1/Q2)
> As shown in https://ibb.co/N6LccrRR, UVR preserves image quality over a broad range (0.3–0.65), while safety effectiveness improves monotonically as $\tau$ decreases.
> This indicates low sensitivity to $\tau$ on image quality, allowing a fixed value to achieve strong safety with minimal quality trade-off in practice, without exhaustive per-concept tuning.
> The selection of $\tau$ is mainly related to differences in the model’s internal representations of concepts; for instance, **general concepts** like nude and blood associating with multiple attributes tend to yield higher attention scores and favor larger $\tau$ (0.5–0.6), whereas **concrete concepts** (Pikachu, weapon) work well with lower $\tau$ (0.3–0.4).
>
> # R2: Generalization of UVR to FLUX.1-schnell (W2)
> We further evaluate UVR on FLUX.1-schnell and observe that the attention dynamics analyzed in Fig.3 (based on -dev) remain consistent, without requiring re-collection of anchors.
> As shown in https://ibb.co/bg98K7Vr, the harm ratio is reduced from 42.89% to 14.69%, while CLIP scores remain stable (31.50 → 31.51).
> UVR adapts effectively due to two factors: (i) UVR’s intervention is defined as a proportion of total denoising steps, allowing it to naturally adapt to compressed temporal schedules. (ii) -schnell is distilled from -dev, leading to aligned semantic representations that enable direct transfer of anchors.
>
> # R3: Scalability to massive and highly abstract unsafe concepts & Automated VLM-based Anchor Construction (Q1)
> For **massive concept erasure (e.g., 100+ celebrities or styles)**, these concepts are often highly abstract and diverse, making them difficult to represent with a single anchor.
> In such cases, the number of anchors scales linearly with the number of concepts $N$ ($\mathcal{O}(N)$) and exploring VLM-assisted adaptive anchor construction (e.g., CLIP-based retrieval or grounded LVLMs) for open-set concepts is a promising direction.
>
> However, we consider such tasks to be primarily text-centric and well addressed by approaches such as MACE [1], whereas the proposed **Unified Visual Safety Regulator (UVR)** focuses on visually grounded harmfulness via visual anchors.
> Theoretically, **a variant of UVR** could be implemented by replacing visual anchors with text-based anchors, where the additional computational cost for information flow modulation scales as $\mathcal{O}(KN)$, with $K$ denoting the cost per attention computation (constant for text-based anchors).
> Overall, this remains a challenging yet promising direction for our future work.
>
> We further consider another class of tasks that aim to **suppress biased concepts from massive and abstract concepts** via gender bias mitigation. We extend UVR to 75 concepts from the MMDT benchmark [2] under T2I settings. Specifically:
> - We perform gender-specific anchor collection and modulate information flow to balance underrepresented groups using auxiliary text vectors.
> - We observe that *attention dynamics generalize* to such task: **attention score residuals** between groups at early steps reveal bias—e.g., 0.119 for Nurse (female-biased), –0.107 for Software Engineer (male-biased), and 0.014 for Doctor (unbiased), which enables targeted regulation via information flow modulation.
> - As shown in https://ibb.co/VYhyRjj9, on MMDT benchmark (62/13 concepts for Occupation/Education), UVR **reduces Group Unfairness from –0.429/–0.556 to –0.048/–0.071 (closer to 0 is better)**, demonstrating strong scalability and generalization of UVR on massive concepts to large and abstract concept spaces.
>
> [1]MACE: Mass Concept Erasure in Diffusion Models, CVPR24
>
> [2]MMDT: Decoding the Trustworthiness and Safety of Multimodal Foundation Models, ICLR25
>
> ---
>
> *Finally*, we sincerely appreciate the reviewer’s constructive and insightful feedback, which helps improve the quality of the paper and ensures it is safe, respectful, and accessible to all readers. We apologize for any discomfort caused by the uncensored images in the initial submission; we will thoroughly revise the presentation of figures and carefully blur and censor all explicit content (including Fig.6, 10, 11, and 13).
>
> Best regards,
>
> Authors

---

> > ### Author Rebuttal · Reviewer_xnpP · 2026-04-03
> >
> > Thanks for the rebuttal. The author's explanation of Practical Scalability clarified my concerns, and the relevant experiments were conducted in FLUX.1-schnell. Thus I maintained my positive score.

---

> > > ### Author Response · Authors · 2026-04-03
> > >
> > > Dear Reviewer xnpP,
> > >
> > > We are sincerely grateful for your thoughtful review and constructive suggestions. Your feedback has been very helpful in strengthening both the technical presentation and the overall completeness of our work. We are glad that our rebuttal has fully addressed your concerns, and we will carefully incorporate your suggestions into the final version of the manuscript, including additional parameter studies, generalization and scalability experiments, and related discussion.
> > >
> > > Best regards,
> > >
> > > Authors

---

### Official Review · Reviewer_D9Eg · 2026-03-12

**Soundness:** 3
**Presentation:** 3
**Significance:** 3
**Originality:** 3
**Overall Recommendation:** 4
**Confidence:** 3

**Summary:**

This work proposes a training-free safe generation framework to mitigate unsafe content in generated images. It performs layerwise and timestep-wise information flow through MM-attention to identify stages that contribute to unsafe semantics and utilizes a visual safety regulator to modulate the targeted attention to prevent the information flow. It achieves significant performance for image synthesis as well as editing tasks without compromising the image quality.

**Compliance With Llm Reviewing Policy:**

Affirmed.

**Final Justification:**

My concerns are fully resolved. So I would like to maintain my initial score and leaning towards acceptance of the paper.

**Key Questions For Authors:**

1. Need more details on unsafe anchors.

2. Technical clarification on targeted attention modulation.

**Limitations:**

Yes.

**Strengths And Weaknesses:**

**Strengths**

1. This is a training-free framework to handle safety for image synthesis as well as editing tasks in DiT architectures.

2. The layerwise and timestep-wise analysis of MM-attention is quite impactful in understanding the information flow of unsafe semantics in the output patches.

3. The visual safety regulator provides explicit restrictions on harmful information via targeted attention modulation.

**Weaknesses**

1. The main text states $\lambda_0$ > 1 for benign enhancement (confirmed by $\lambda_0 = 1.05$ in Appendix C.1), but Algorithm 2 declares 0 < $\lambda_0$ < 1. These are directly contradictory.

2. Eq. 2 collects all unsafe patches exhaustively, while Algorithm 1 introduces a budget K and an undefined SelectTopK(·) selection function. These describe incompatible procedures.

3. It is not clearly described how the unsafe anchors are obtained. Are they generated via prompts?

4. What is target attention modulation in the UVR step? Is it performed to zero out the attention weights? In which layers and diffusion is this applied? Elaborate the early stage.

---

> ### Author Rebuttal · Authors · 2026-03-31
>
> We appreciate the time and careful reading from the reviewer. We hope the clarifications provided below help further clarify our work.
> # R1: Explanation on the Definition of $\lambda0$ in Attention Modulation (W1/Q2)
>
> We thank the reviewer for pointing out this typographical error. To resolve it, we will consistently use $\delta \lambda_0 = 0.05$ in Algorithm 2, corresponding to $\lambda_0 = 1.05$ as reported in Appendix C.1 and in Main text.
> # R2: Details on Unsafe Anchor Selection (W2/Q1/Q2)
> We clarify that Main.Eq.2 and Algorithm 1 in Main.Appendix.B are consistent. Main.Eq.2 defines the collection process for unsafe image anchors, while Algorithm 1 introduces a practical budget K to ensure computational efficiency during collection. Specifically, the SelectTopK(·) function in Algorithm 1 evaluates and filters candidate anchors based on their attention score differences between unsafe and safe regions, efficiently capturing the most discriminative patches from the exhaustive candidate space as anchors.
> We thank the reviewer for raising this point. We will revise the description to ensure consistency across the main text and appendix.
>
> # R3: Clarification on Collection of unsafe anchors. (W3/Q2)
> Unsafe anchors are obtained by prompting the model with unsafe inputs (either text or reference images) to generate the target concept, thereby accurately capturing the model’s learned visual-linguistic representations. Specifically, we first execute a generation pass using the unsafe input and save the target image embeddings from the MMDiT at the final denoising timestep. Next, we apply GroundingSAM to the generated image to obtain the exact spatial coordinates of the unsafe visual patches. Finally, using these coordinates, we extract the corresponding patch embeddings from the saved final-timestep features as unsafe anchors.
> # R4: Details on target attention modulation (mechanism, timing, and layers) (W4/Q2)
> UVR’s attention regulation aims to recover and generate safe content by recognizing and blocking unsafe information flow in MMDiT before unsafe features form, rather than simply zeroing out attention weights.
> Specifically, for a located unsafe index $i$, the regulation operates as follows:
> $$O_t(i)=(1-\alpha_t)\sum_j\left[\mathcal{S}\left(\lambda_t(i,j)\cdot a_t(i,j)\right)V_t(j)\right]+\alpha_t\boldsymbol{\epsilon}$$
> This dual-action mechanism consists of:
> - **Attention modulate($\lambda$):** diminishes specific unsafe attention weights to prevent further unsafe information from flowing into position $i$.
> - **Noise perburbation($\alpha$):** Injects Gaussian noise $\boldsymbol{\epsilon}$ to disrupt unsafe representations the model has already begun to learn.
> To maintain high image quality, noise perturbation ($\alpha$) is limited to the first 10 of the entire denoising steps (28), while attention modulation ($\lambda$) spans the first 23 steps, encouraging the model to rely on its internal safe generation priors rather than the input condition.
> Visual anchor localization is applied specifically to **single blocks** where the most frequent cross-modal (and unsafe) information exchange occurs, whereas noise perturbation and attention modulation are applied **across all blocks** to ensure effective disruption.

---

> > ### Author Rebuttal · Reviewer_D9Eg · 2026-04-03
> >
> > I appreciate the rebuttal. My concerns have been resolved.

---

> > > ### Author Response · Authors · 2026-04-03
> > >
> > > Dear Reviewer D9Eg,
> > >
> > > We sincerely appreciate the time and effort you devoted to reviewing our submission. Your careful reading and constructive feedback have greatly helped us improve the presentation of the paper. We are glad that our rebuttal has fully addressed your concerns, and we will carefully revise the manuscript accordingly.
> > >
> > > Best regards,
> > >
> > > Authors

---

### Official Review · Reviewer_mau3 · 2026-03-12

**Soundness:** 3
**Presentation:** 3
**Significance:** 3
**Originality:** 3
**Overall Recommendation:** 4
**Confidence:** 3

**Summary:**

This work proposes UVR, a training-free framework for suppressing unsafe content in MM-DiTs for both T2I and I2I tasks. The key insight is a task-agnostic "semantic priming phase" in early diffusion steps where unsafe semantics are injected into output patches; UVR localizes these regions via anchor patch embeddings and suppresses their propagation through adaptive attention modulation.

**Compliance With Llm Reviewing Policy:**

Affirmed.

**Key Questions For Authors:**

1. How do anchor set size and diversity affect localization precision? Specifically, how many samples are needed to build an effective anchor set for a given unsafe concept? Have you conducted a sensitivity analysis on anchor quantity? Does increasing the anchor count reduce FPR?

2. What is the inference time overhead of UVR? Compared to standard FLUX inference, how much additional latency does UVR add per image?

3. The threshold tau currently requires per-concept manual tuning. Is it feasible to develop an adaptive thresholding mechanism? For example, to automatically determine tau based on statistical properties of the anchor distribution?

4. The paper focuses on visually explicit unsafe concepts (nudity, IP characters, weapons). For more abstract unsafe concepts (e.g., racial bias, gender stereotypes), does patch-level localization remain effective? These concepts may not correspond to specific spatial regions, potentially requiring a fundamentally different approach.

**Limitations:**

Please refer to the weakness and question section.

**Strengths And Weaknesses:**

# Strengths

- **Addresses a genuine safety gap**. Existing safe generation methods focus on T2I or U-Net architectures, leaving I2I editing vulnerable， so that users can bypass text-level safety filters by supplying unsafe reference images. UVR's unified treatment of both tasks within MM-DiTs fills a real and underexplored need.

- **Insightful attention dynamics analysis**. The temporal analysis in Section 3 reveals that T2I unsafe semantics concentrate in early denoising steps, while I2I reference interference persists throughout, providing actionable architectural understanding.

- **Elegant and practical design**. Localizing unsafe content on output image patches rather than on inputs is a key insight: since unsafe signals from both text and reference images ultimately manifest in output patches, a single mechanism handles both T2I and I2I without task-specific logic. The evaluation is comprehensive, and the training-free design enables straightforward deployment.

# Weaknesses

- **Limited anchor generalization**. Unsafe anchors are constructed from predefined prompts and Grounded-SAM masks, creating a dependency on pre-specifying unsafe concepts that may not generalize to emerging or culturally nuanced content. The paper does not discuss how many samples are needed per concept, and the threshold τ requires manual tuning per category (0.6 for nudity, 0.35 for Pikachu, 0.3 for weapons) with no adaptive mechanism proposed.

- **Narrow comparison and missing efficiency analysis**. All baselines require training or fine-tuning, making it unclear whether UVR's lower erasure rates reflect an inherent ceiling of training-free approaches. A comparison against a simple post-hoc filter (e.g., NudeNet) would better contextualize the contribution. Inference overhead, i.e., UVR computes anchor similarity and attention modulation at every denoising step, is not reported.

---

> ### Author Rebuttal · Authors · 2026-03-31
>
> We sincerely thank the reviewer for the detailed feedback and for recognizing the novelty, practical value, and importance of our work in addressing an underexplored safety gap. We hope the clarifications below further address the concerns and improve the clarity of the paper.
>
> # R1: Ablation study on Anchor Set (Q1/W1)
> We show that using only 1–2 images to construct the anchor set is sufficient to accurately localize unsafe regions and perform effective regulation that generalizes to unseen harmful content.
> As shown in the table, when using 2 images, the CLIP score reaches 31.32 (vs. 31.31 for FLUX) with low false positives, while the Harm Ratio is reduced to 9.3% (vs. 43.2% for FLUX). Increasing the number of images yields consistent but marginal improvements (e.g., 8 images: CLIP 31.33, Harm 8.9%).
> Considering the additional computational cost of larger anchor sets (29/125 anchors for 2/8 images), we adopt 2 images as a practical trade-off between efficiency and performance.
>
> | **Number of Images**          | 1     | **2**         | 3     | 4     | 5     | 6     | 7     | 8     |
> | ------------------------------------ | ----- | --------- | ----- | ----- | ----- | ----- | ----- | ----- |
> | **CLIP Score (Utility $\uparrow$)**  | 31.31 | 31.32 | 31.32 | 31.32 | 31.32 | 31.33 | 31.32 | 31.33 |
> | **Harm Ratio (Safety $\downarrow$)** | 9.4%  | 9.3%  | 9.3%  | 9.2%  | 9.1%  | 9.2%  | 9.1%  | 8.9%  |
>
> # R2: Inference Efficiency (Q2/W2)
> We provide an efficiency analysis of latency and inference overhead for UVR and baselines in https://ibb.co/zhknXSPG. All metrics are evaluated on the same platform. Compared with fine-tuning-based methods (ESD, EA, DES) and editing-based methods (UCE), UVR incurs only a slight memory overhead (~3 MB) relative to FLUX, and achieves lower latency (25s) than the training-free method SLD (31s), as UVR intervenes only in early denoising steps (SLD requires two forward passes per step).
> Specifically, the average time for anchor similarity computation and attention modulation is 2s/4s for each image. Specifically, the average time for anchor similarity computation and attention modulation is 2s and 4s per image, respectively.
>
> # R3: Comparison with NudeNet (W2)
> To better contextualize our contribution, we present a comparative study of unsafe detection between NudeNet and UVR in https://ibb.co/2YcmQH2x. We introduce the average attention score to unsafe anchors as a proxy metric for UVR’s detection confidence (UVR Score, range [0,1]) and compare it with the NudeNet score (confidence of detected unsafe classes in [0,1]).
> The UVR Score and NudeNet score show strong consistency on unsafe images, indicating that UVR has the potential to serve as an internal detector.
> Moreover, as an internal regulation framework, UVR is orthogonal to post-hoc filtering: it improves safety during generation while preserving user experience through high-quality outputs.
>
>
> # R4: Feasibility of Adaptive Thresholding $\tau$ (Q3/W1)
> - **Clarification on $\tau$:** Our ablation study in https://ibb.co/N6LccrRR shows that UVR does not require precise per-concept tuning, preserving image quality over a broad range (0.3–0.65), while safety effectiveness decreases monotonically as $\tau$ increases.
> - **Adaptive Mechanism**: The patterns exhibited by $\tau$ across different conceptual categories support the feasibility of adaptive thresholding. For instance, **general concepts (e.g., nudity, blood)**, which are associated with multiple attributes, tend to yield higher attention scores and favor larger $\tau$ (0.5–0.6), whereas **concrete concepts (e.g., Pikachu, weapon)** work well with lower $\tau$ (0.3–0.4).
>
> # R5: Extending UVR to Mitigate Gender Stereotypes (Q4)
> Patch-level localization remains effective for debiasing tasks with a slightly different formulation, yet can be implemented within the same framework. The **key difference is that**, *rather than erasing specific concepts, debiasing aims to balance sensitive attributes (e.g., gender) within socially meaningful concepts (e.g., occupations)*.
> We extend UVR to mitigate gender stereotypes in T2I generation as follows:
> - We perform *gender-specific anchor collection* (used for localization via attention score residuals) and *modulate information flow* to balance underrepresented groups using auxiliary text vectors.
> - We observe that *attention dynamics generalize to this setting*: attention score residuals between groups effectively reveal bias—for example, 0.119 for Nurse (female-biased), –0.107 for Software Engineer (male-biased), and 0.014 for Doctor (unbiased)—enabling targeted regulation through information flow modulation.
> - As shown in https://ibb.co/VYhyRjj9, on MMDT benchmark[1] (62/13 concepts for Occupation/Education), UVR **reduces Group Unfairness from –0.429/–0.556 to –0.048/–0.071 (closer to zero indicates better fairness)**, demonstrating strong scalability to more abstract and diverse undesirable concepts.

---

> > ### Author Rebuttal · Reviewer_mau3 · 2026-04-02
> >
> > I appreciate the rebuttal. My concerns have been resolved.

---

> > > ### Author Response · Authors · 2026-04-03
> > >
> > > Dear Reviewer mau3,
> > >
> > > Thank you very much for your thorough review and constructive feedback, which have been highly valuable in improving the quality and clarity of our paper. We are glad that our rebuttal has fully addressed your concerns, and we will carefully incorporate your suggestions on ablation and parameter studies, efficiency analysis, and extensive fairness experiments into the final version of the manuscript.
> > >
> > > Best regards,
> > >
> > > Authors

---

### Review · Ethics_Reviewer_C8uq · 2026-03-27

**Recommendation:** Remediation action needed

**Ethics Issue:**

This project describes a useful intervention to help reduce production of harmful content. However, as other reviewers noted, it does not seem necessary to contain unblurred nudes in order to illustrate some of the concepts. While the warning at the start of the article is helpful, I would also recommend blurring images unless it is necessary to make a specific technical point. It also seems worth noting (although not in immediate need of remedy) that all of the images of nudity are of women, producing a net effect that may be differentially impactful in associating women with nudity and potentially contributing to a hostile environment to women researching in the field.

It is also worth discussing in the impact statement that the kinds of mitigation to which this approach is suited are those where context-sensitivity is not a consideration. While it may be important that some kinds of images be wholly impermissible to generate via a given system, in other cases, details of context may make a significant difference, and this intervention is not equipped to handle them. Making the narrower scope of this project clear can help make its ethical application more apparent.

**Remediation Action:**

Blur explicit nudes, since they do not appear to be necessary to support the discussion and explanation, consider varying up examples to avoid reinforcing gendered implication, and clarify narrower scope of intervention (most appropriate for blanket-banned content rather than context-sensitive material) in the impact statement.

---

### Decision · Program_Chairs · 2026-04-30

**Decision:**

Accept (regular)

**Comment:**

The reviewers gave three "Weak Accept" ratings and one "Weak Reject" rating, indicating a general consensus leaning towards acceptance. This paper proposes a training-free framework that mitigates unsafe content generation in multimodal diffusion transformers across both text-to-image and image-to-image tasks. The reviewers appreciate the insightful analysis of attention dynamics and the unified design that successfully addresses a genuine safety vulnerability without requiring model fine-tuning. Although initial concerns were raised regarding the framework's reliance on predefined unsafe anchors and external visual grounding models, the authors provided a rebuttal to address these concerns. The authors are required to address the highlighted ethical concerns by blurring explicit images and mitigating potential gender biases in their revised manuscript. Overall, the AC is leaning to accept this paper.